

# CardioTF, a database of deconstructing transcriptional circuits in the heart system

Yisong Zhen

State Key Laboratory of Cardiovascular Disease, Fuwai Hospital, National Center for Cardiovascular Diseases, Chinese Academy of Medical Sciences and Peking Union Medical College, Beijing, China

Corresponding author
Yisong Zhen,
zhenyisong@fuwaihospital.org

## ABSTRACT

**Background:** Information on cardiovascular gene transcription is fragmented and far behind the present requirements of the systems biology field. To create a comprehensive source of data for cardiovascular gene regulation and to facilitate a deeper understanding of genomic data, the CardioTF database was constructed. The purpose of this database is to collate information on cardiovascular transcription factors (TFs), position weight matrices (PWMs), and enhancer sequences discovered using the ChIP-seq method.

**Methods:** The Naïve-Bayes algorithm was used to classify literature and identify all PubMed abstracts on cardiovascular development. The natural language learning tool GNAT was then used to identify corresponding gene names embedded within these abstracts. Local Perl scripts were used to integrate and dump data from public databases into the MariaDB management system (MySQL). In-house R scripts were written to analyze and visualize the results.

**Results:** Known cardiovascular TFs from humans and human homologs from fly, *Ciona*, zebrafish, frog, chicken, and mouse were identified and deposited in the database. PWMs from Jaspar, hPDI, and UniPROBE databases were deposited in the database and can be retrieved using their corresponding TF names. Gene enhancer regions from various sources of ChIP-seq data were deposited into the database and were able to be visualized by graphical output. Besides biocuration, mouse homologs of the 81 core cardiac TFs were selected using a Naïve-Bayes approach and then by intersecting four independent data sources: RNA profiling, expert annotation, PubMed abstracts and phenotype.

**Discussion:** The CardioTF database can be used as a portal to construct transcriptional network of cardiac development.

**Availability and Implementation:** Database URL: http://www.cardiosignal.org/database/cardiotf.html.

## INTRODUCTION

Heart disease is a leading cause of morbidity and mortality in both infants and adults (*van der Linde et al., 2011*; *Celermajer et al., 2012*). Insights into the cause of congenital heart diseases (CHDs) have led to the identification of mutations in essential cardiac transcription factors (TFs) (*McCulley & Black, 2012*). At the opposite end of the temporal

spectrum, some cases of adult cardiac disease have been traced to variation in gene regulatory sequences (*Smith & Newton-Cheh, 2015*). Thus, knowledge of TFs, their downstream targets, and the regulatory genomic sequences involved in the heart development will enhance our understanding of heart disease.

Although the vast amounts of data generated by high-throughput technologies are archived in databases such as ArrayExpress or GEO of NCBI (*Parkinson et al., 2011*; *Barrett et al., 2013*), they do not contain cohesive knowledge and lack expert annotation. In addition, the field of cardiac development has experienced accelerated growth that can be attributed to the use of various animal models. However, to our present understanding, few efforts have been made to create a database which collects cardiac transcriptional information across species, thereby limiting the benefits from an evolutionary perspective to study heart development.

At present, two branching efforts have been made to archive and analyze the data. One is to construct small scale databases, like BloodChIP or CistromeMap, which are dedicated to collecting specific types of data (*Chacon et al., 2014*; *Qin et al., 2012*). The other approach is to establish a number of consortia, like ENCODE, modENCODE, and Epigenomics Roadmap, which are created to generate huge amounts of raw data and archive them (*Harrow et al., 2012*; *Celniker et al., 2009*; *Romanoski et al., 2015*). In addition to these projects, analysis and visualization software are valuable resources that lead to deeper understanding of the data, and facilitate the generation of novel hypotheses. Central databases, like Ensembl and UCSC also have search functions which allow browsing of the results generated by the consortia mentioned above (*Mangan et al., 2014*; *Flicek et al., 2013*). However, there are currently few databases committed exclusively to cardiovascular development (*Djordjevic et al., 2014*). This prompted us to combine information about TFs, position weight matrices (PWMs), and ChIP-seq results and create a one-stop site for information on cardiovascular development, thus facilitating systems biology studies in transcriptional network regulation (*Blais & Dynlacht, 2005*).

CardioTF was therefore constructed to capture all transcriptional information relating to cardiovascular development. As a biocuration project, it documents TFs, PWM files and enhancers across species, including fly, *Ciona*, fish, frog, chicken, mouse and human. It also implements a search engine to query this information on the fly. In addition to the data-mining effort, core cardiac TFs are identified using Naïve-Bayes approach, which can be used as a roadmap alongside with further annotation for enhancers to generate gene regulatory network of heart development.

## MATERIALS AND METHODS

### The project's code and data for reproducible research

All the Perl scripts and R codes were uploaded to GitHub (https://github.com/zhenyisong/). The raw data including Weinstein meeting abstracts (positive_test_data plus positive_training_data), negative dataset (negative_test_data plus negative_training_data) (in zip format), intermediary files, which contain cross-validation results as well as other public data were uploaded to the CardioTF database server (http://www.cardiosignal.org/download/download.html). These raw data and source codes can be used to verify the findings.

## Comprehensive collection and annotation of cardiac TFs

Cardiac TFs were previously defined as regulators of cardiac gene expression, which can impact the process of heart development, particularly the initiation and maintenance of the myocardium (*Zhen et al., 2007*). In the CardioSignal database, efforts were also made to collate the cardiac specific enhancers which drive gene expression in cardiomyocytes. At that time, cardiac specific transcriptional factors were defined as genes that regulated expression of genes in the myocardium. During development, the heart consists of three layers: the myocardium, epicardium and endocardium (*Moorman & Christoffels, 2003*; *Fishman & Chien, 1997*). Additionally, at least four heart-specific cell lineages have been characterized, including cardiomyocytes, endothelial cells, epicardial cells, and fibroblasts, the latter is derived mainly from epicardial cells through the epithelial to mesenchymal transition (EMT) (*Evans et al., 2010*; *Moore-Morris et al., 2016*). By definition, cardiac TFs themselves should be involved in the steps of specification, determination, patterning, and differentiation that will result in a heart fate. In our CardioTF database we collated, cardiac TFs which are expressed in all layers of heart. The goal of the CardioSignal database was to use a machine learning approach to find cardiac enhancers at the genome scale. In contrast, the CardioTF database is constructed to study systems biology of transcription regulation. The cross-talk between different layers will also be explored using this platform. In the initial screen, we identified human TFs from previously published annotations (*Wingender, Schoeps & Dönitz, 2013*), hence it was named Wingender's annotation set. This dataset is comprehensive in annotating human TFs. We used these human TFs, excluding human-specific TFs, as a reference to search for their homologs in other species, including fly, *Ciona*, zebrafish, chicken, frog and mouse (*Hutson & Kirby, 2007*). Human-specific TFs are defined as genes which have no homologs in the mouse genome. The NCBI HomoloGene database (*NCBI Resource Coordinators, 2015*) was used as a reference to assess homologs between human and mouse/zebrafish. Human homologs from other were retrieved from their central databases, namely, FlyBase (Fly), BirdBase (Chicken), Aniseed (*Ciona*) and Xenbase (Frog) (*Attrill et al., 2016*; *Karpinka et al., 2015*; *Schmidt et al., 2008*; *Tassy et al., 2010*). We also documented the expression status for mouse TFs from four independent sources which included annotation from the Cardiovascular Gene Ontology Annotation Initiative (*Khodiyar et al., 2011*), Mouse Genome Database (MGI) genes with cardiovascular phenotypes (*Blake et al., 2014*), PubMed abstract parsing results and RNA expression profiling results.

## TFs from PubMed abstract parsing

The Weinstein Cardiovascular Conference provides a platform for talks and posters on all aspects of heart development and congenital heart disease. The Weinstein meeting abstracts were extracted from the meeting abstract book from 2010 to 2013 by hand. As the positive group, this data set included 954 abstracts. We assumed that Weinstein-like abstracts deposited in PubMed are all from the cardiovascular community and focus on cardiovascular development. Abstracts from the negative group were from

non-heart related journals, which were manually selected from the PMC Open Access Subset at NCBI. To choose the negative control journals, the following criteria were set. First, well-known cardiovascular journals were excluded, such as "Circulation" and "Circulation Research." Second, journals without key words "heart" or "cardiac" or "cardiovascular" in their title were selected. Third, journals which are dedicated to the study of other organs or diseases, for example, "Neuron" or "Cancer" were selected. Fourth, other journal which are unlikely to publish articles about cardiovascular development and related topics, such as journals about plants or viruses, were selected. Journals in the negative group contained research from across kingdoms and topics obviously in other fields, such as "Sleep_Disord" or "Toxicology." All journal names in the negative group were saved in a file and uploaded onto the cardioTF server (negative_set.journal.txt). The negative group includes 57,080 abstracts. We split the data (positive and negative groups) into a training (80%) and test (20%) set. The Naïve-Bayes module from The Comprehensive Perl Archive Network (CPAN) was used with a local Perl script to classify Weinstein-like abstracts. We used the training set and adopted the $5 \times 2$ cross-validation proposed by *Dietterich (1998)* to train and validate the data. The parameter (the cutoff to decide whether an abstract is a true Weinstein-like abstract) was selected based on average predictive performance which resulted in a classification accuracy (ACC) of 0.99. A wrapper function was implemented to parse the abstracts and calculate the word frequencies. This function called two Perl modules (Lingua::EN:Splitter and Lingua::EN::StopWords) to extract words and perform text analysis. The word frequency alone was forwarded to the algorithm. The withheld test set using the optimized parameter was then used to assess the algorithm's final performance. All publication abstracts from 2008 to 2013 were downloaded to the local environment and analyzed by the algorithm. We targeted journals which had at least six publications classified as Weinstein-like abstracts in the six-year period (annual publication rate is $\geq 1$). Then all abstracts from the targeted journal were downloaded. This process was repeated for all journals that met the criteria. The selected abstracts were then processed by GNAT (*Hakenberg et al., 2008*) using its default script (test100. sh) to recognize the mouse gene name. The PMID was recorded when the gene name matched the name in the curated mouse TF set.

## RNA expression profiling data procession

Affymetrix data (GSE1479) were processed by R using the MAS5 algorithm which provides a present call for each gene (see the script ExtractAffy.R at Github) Gene expression status was defined as "on" if the gene was expressed in any microarray at selected developmental stages and had a present call. RNA-seq data were re-analyzed using the recommended protocol (all raw data identifiers can be retrieved from Table S2) (*Trapnell et al., 2012*). Briefly, pre-processing software (FastQC) was used to estimate the read length of raw data. If read length is above 50 bp, Bowtie2 was used. Otherwise, Bowtie was used. Mm10 and hg19 are the genome builds used by UCSC. Index and annotation files for Bowtie2/Bowtie were downloaded from Illumina iGenomes project. Genome sequences from UCSC are repeat-masked with lower-case characters.

Any gene with an FPKM value greater than one was defined as expressed and this threshold was empirically set although justifiable.

## Depositing PWM files

The gene symbol was used as the unique identifier to link the original database ID to our local database primary key. A local Perl script was written to change the format to the TRANSFAC style, which was used by our in-house CardioSignalScan program (*Zhen et al., 2007*). PWM files were collected from Jaspar, UniPROBE and hPDI databases (*Mathelier et al., 2014*; *Hume et al., 2015*; *Xie et al., 2010*). Users can retrieve their annotations by directing them to the respective database. All the PWMs from those three sources for each TF were deposited in the database. We currently do not use the program implemented by the Zhang lab (*Schones, Sumazin & Zhang, 2005*) to check the similarity of PWMs and reduce the redundancy in the collection of PWM files.

## Orthologs of TFs from model systems

NCBI has its own gene orthologs that were identified using unpublished algorithm (*Altenhoff & Dessimoz, 2009*). TFs from mouse, human and zebrafish are annotated by NCBI Homologene (*NCBI Resource Coordinators, 2015*). Frog, chicken and *Ciona* TF homolog annotations were downloaded from their central databases including Xenbase, BirdBase and ANISEED. Fly TFs, which have counterparts in the human proteome, were annotated by the Inparanoid system (*Sonnhammer & Östlund, 2015*). Each TF collected in the database was assigned one treeID on the basis of its human counterpart. The treeID is equivalent to a TF family by the recommendation of TFClass (*Wingender, Schoeps & Dönitz, 2013*).

## Enhancer curation: TF-ChIP and Histone-ChIP data processing

Raw ChIP-seq data were recruited based upon two criteria: first, whether the source of tissue or cells is from heart or heart progenitor derived cells; second, the DNA-binding protein for the ChIP assay should be pan-enhancer markers or heart lineage specific TFs. In the latter case, the core heart TFs were proposed in our screening procedure. Enhancer regions were defined by ChIP-seq signals. We assume that pan-enhancer markers, like H3K4me1 or H3K27ac (*Shen et al., 2012*), or lineage specific markers, like GATA4 or MEF2C (*He et al., 2011*) will delineate true enhancer regions, although these collections will produce some false positive records. Peak calling was performed using the recommended pipeline (*Bailey et al., 2013*). In brief, sequencing reads were aligned to the mm10/hg19 reference genome using Bowtie/Bowtie2. Mm10/hg19 represents the genome build assigned by UCSC. Index files for mm10/hg19 were downloaded from the iGenome project. MACS1.4.2 was used to process all the ChIP-seq data. The default cutoff for the p-value was 1e-05. This default value was used in all ChIP-seq analysis. This protocol was adapted from published literature (*Feng et al., 2012*).

Bowtie call

bowtie -m 2 -S -q -p 8

Peak calling was performed using the MACS peak calling algorithm.

MACS call linux command

macs14 -t ERR231646.bam -c ERR231653.bam -g mm -n sham_Anti_H3k9ac.

A Torque job script was written to submit the job to the supercomputer. After that, the format transformation was performed:

samtools view -bS -o tbx20_positive.bam positive_tbx20.sam

When possible, the control files were merged:

samtools merge out.bam in_1.bam in_2.bam in_3.bam.

After MASC analysis was completed, the annotatePeaks.pl was run in HOMER (*Heinz et al., 2010*) to parse the bed file from the MACS output. Then the parsed results were dumped into the MySQL table. Public identifiers for the raw data can be retrieved from Table S2 and ChIP-seq experimental information has been recorded in the MySQL table "ChIPExpAssay."

## Recognition of transcription factor binding sites (TFBSs) in enhancer

CardioSignalScan was previously implemented to identify transcription factor binding sites (*Zhen et al., 2007*). However, this local program (see cardiophylo.pl in GitHub) is brute-force solution which consumes computational time with linear complexity ($O(mn)$). In the Big O notation, m is the column length of the matrix and n is the length of the input DNA string. Therefore, it is unrealistic to scan sequences longer than 3,000 bp with this local program. This prompted us to choose MOODS (*Korhonen et al., 2009*) instead, which reduces the computational time proportionally to PWMs length ($O(m)$). A wrapper module was written to calculate the threshold that gauges the match. The cutoff was empirically defined to be 0.75 (range from 0–1 and 1 is most conserved score).

$$\text{threshold} = \text{min\_log\_score} + (\text{max\_log\_score} - \text{min\_log\_score}) * \text{cutoff}$$

This step avoids using p-values to assess the significance of TFBS.

## Gene ontology analysis

DAVID analysis (version 6.7) was performed using the 81 TFs as the input gene list, official gene symbols as the identifiers and the entire mouse gene set as the background. The functional annotation clusters generated by DAVID were identified by TFs (Fig. S2). The classification stringency was set to the default (medium).

## RESULTS

### The database schema

Our database uses the MariaDB, a drop-in replacement for MySQL, as the database management system (DBMS). To address how information will be stored and how the

elements will be related to one another, we used the unified modeling language (UML) to describe the high-level database model (*Ullman & Widom, 2008*). UML was originally developed as a graphical notation for describing software designs in an object-oriented style. It has been extended, and modified and is now a popular notation for describing database designs. Here, we used UML instead of an entity/relationship diagram to design the relational database schema following modeling principles, such as faithfulness, avoiding redundancy, and simplicity counts (*Ullman & Widom, 2008*) (Fig. 1). Where possible, we used a composition distinguished by a line between two classes that ends in a solid diamond at one end. The diamond implies that the label at the end must be 1:1. For example, there is a composition from CardioTFmatrix to CardioTFCenter, which means that every matrix annotation row (PWM related information) belongs to exactly one row in CardioTFCenter (one type of TF may have more PWM records in a CardioTFmatrix table). A 1:1 label at the CardioTFCenter end is implied by a solid black diamond.

## Web interface and search engine

CardioTF is a Perl website implemented using only Perl language to dynamically display the graphical output while querying the database in the backend (Fig. 2). To aid cardiovascular biologists, a search engine was created to allow users to: (1) identify homology information for the queried TF across six species and link to the corresponding central databases outside CardioTF; (2) identify PWM file union of three public databases regarding the queried TF; and (3) identify the enhancer regions revealed by ChIP-seq data of the queried gene. Thus, the database is able to perform the key functions required to construct a transcriptional network of heart development.

## Cardiovascular TFs in the database

Wingender's annotation set (*Wingender, Schoeps & Dönitz, 2013*) was used as a benchmark to recruit TFs across species. The frozen version of this dataset contains 1,564 human TFs. Among them, only 1,513 TFs have corresponding Entrez gene records. Human-specific TFs, defined as those with no orthologs in the mouse genome, were discarded because no model system could be used to verify their function in vivo. This step excluded a further 313 human TFs which have no counterpart in mouse from the homolog annotation. Therefore, 1,200 mouse TFs were collected. Other established animal models for cardiovascular development include fly, *Ciona*, zebrafish, frog, and chicken. TFs from these species were collected if they were homologs to the above mouse TFs. The distribution of TFs from different species is shown in Fig. 3. The expression status of mouse TFs was verified by four independent resources, namely RNA-seq data re-analysis (*Shen et al., 2012*), phenotype annotations from the MGI database (*Blake et al., 2014*), expert recommendation from the UK Cardiovascular Gene Annotation Initiative (*Khodiyar et al., 2011*), and PubMed relevance from classification of Weinstein-like abstracts (see the subsequent section and Table S1).

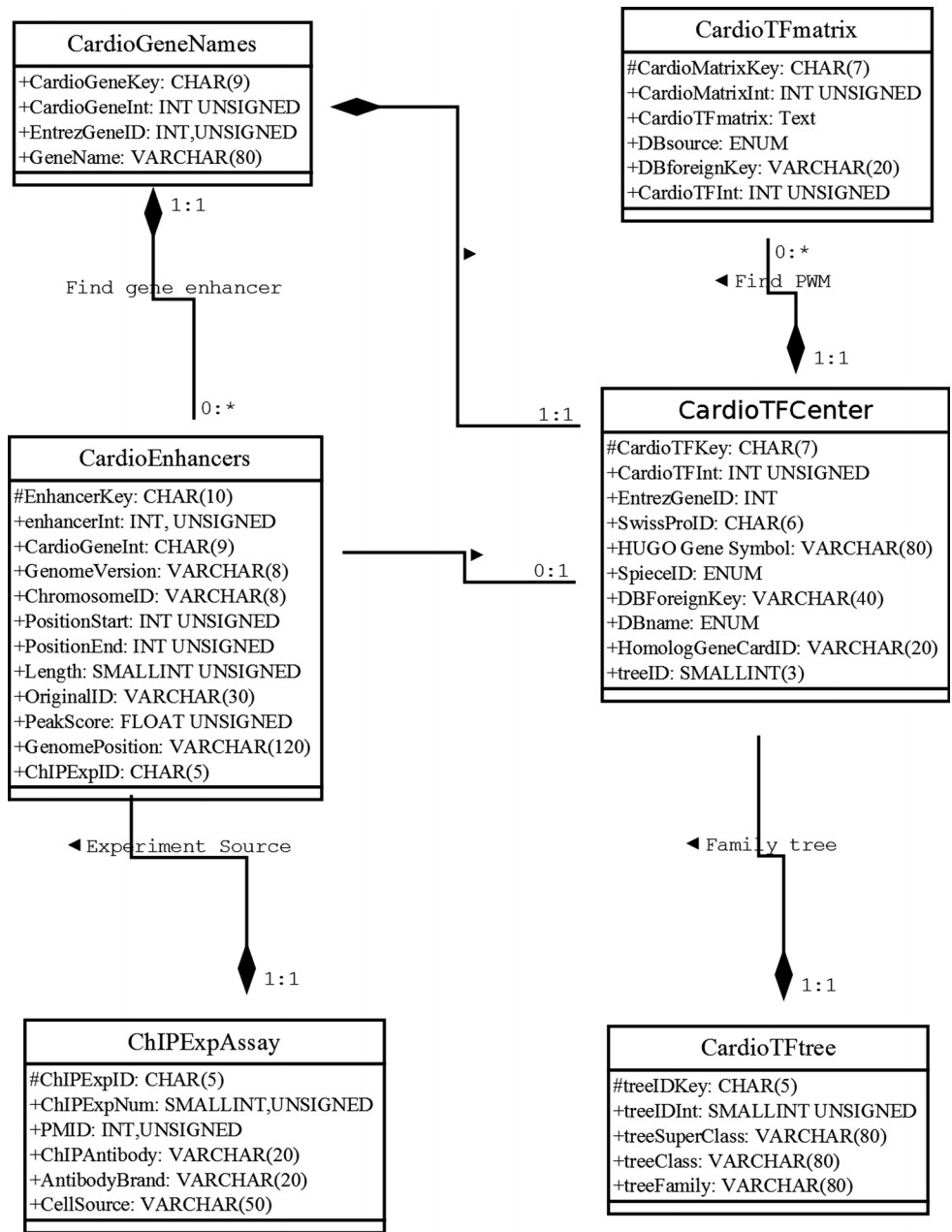

**Figure 1 Unified modeling language diagram for the Cardio-TF database design.** The six boxes represent the six major classes, namely CardioTFmatrix, CardioTFCenter, CardioTree, CardioGeneNames, CardioEnhancer, and ChIPExpAssay. These classes are analogous to entity/relationship sets. Each class has two sections, one for the class name and one for the attributes. The attribute of each class is associated with the type used in MariaDB. The "#" in front of an attribute indicates that it's visibility is "protected," thereby making it a primary key. These classes faithfully represent the real world.

## Weinstein TFs from PubMed analysis

We identified the journals that favored Weinstein-style papers, which were likely contain information on genes involved in cardiovascular development. As expected, after using a Naïve-Bayes method, the journals we identified were among the 30 journals most relevant to developmental biology. Two of the journals (*Circ. Res.* and *J. Mol. Cell. Cardiol.*)

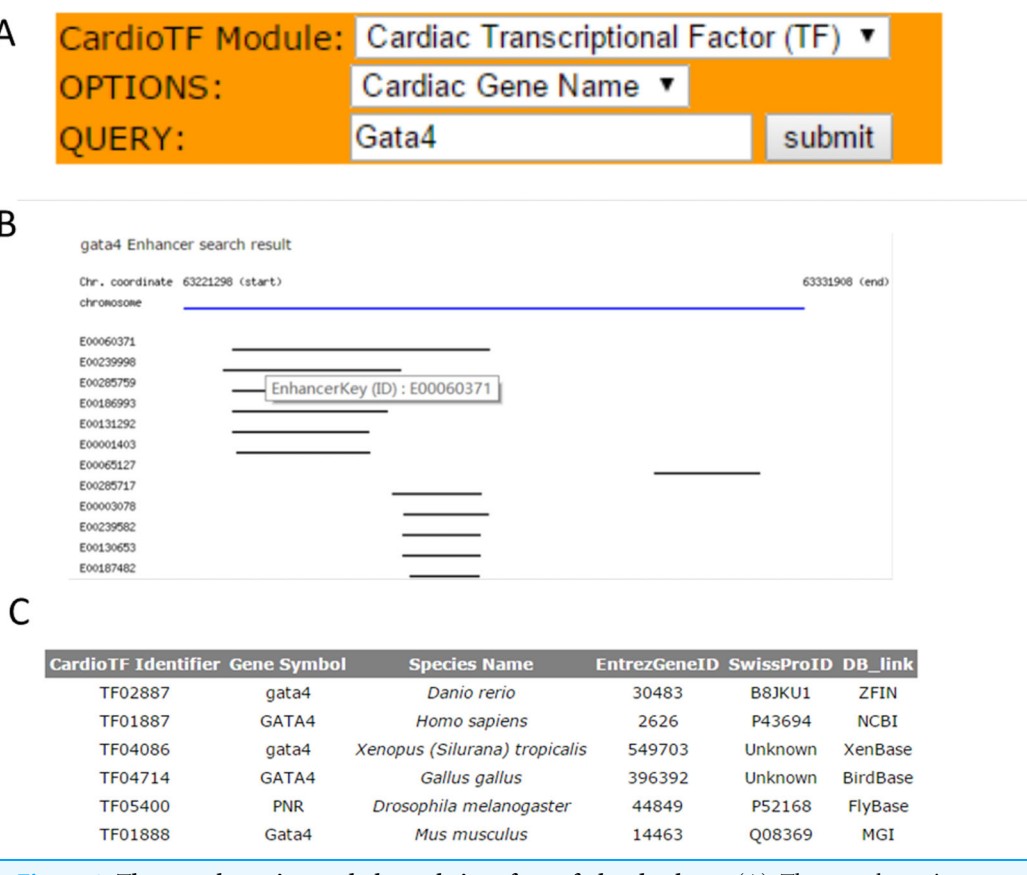

**Figure 2 The search engine and the web interface of the database.** (A) The search engine was implemented to perform three functions: querying TFs, their PWMs and gene enhancers (B) Web graphical output of Gata4 enhancers in mouse. Black lines indicate the enhancer regions found by the ChIP-seq scanning program. These regions are from the same specie, but might be from different experiments. A user can check the experiment inforamtion by clicking the E0000XXX link which represents the primary key (ID) for this enhancer region, thus allowing the user to save it and retrieve the information later. (C) Query results for the GATA4 TF across species. TFs are listed and indexed according to their database identifiers.

obviously publish research specifically in the area of heart system. (Table S1: CardioJournalDistribution). If normalized and ranked by publication rate, the above conclusion still holds true although two different heart journals (*Eur. J. Echocardiogr, Heart Rhythm*) are in the top 30 list (Table S1: CardioJournalDistribution_norm) in this case. We then used GNAT, a tool that recognizes gene names in the literature, to recover all TFs mentioned in Weinstein-style abstracts because we assumed that these TFs are studied by researchers in the cardiovascular community (Figs. 4 and S1; Table S1).

## PWM files collected in database

Public databases for PWM files include UniPROBE, Jaspar, and hPDI, and they provide PWM files for TFs. Jaspar PWM files are curated from the published literatures whereas the other two databases generate PWM files from experiments. Our database integrates these three sources, and the TF PWM can be queried on the basis of the TF name.
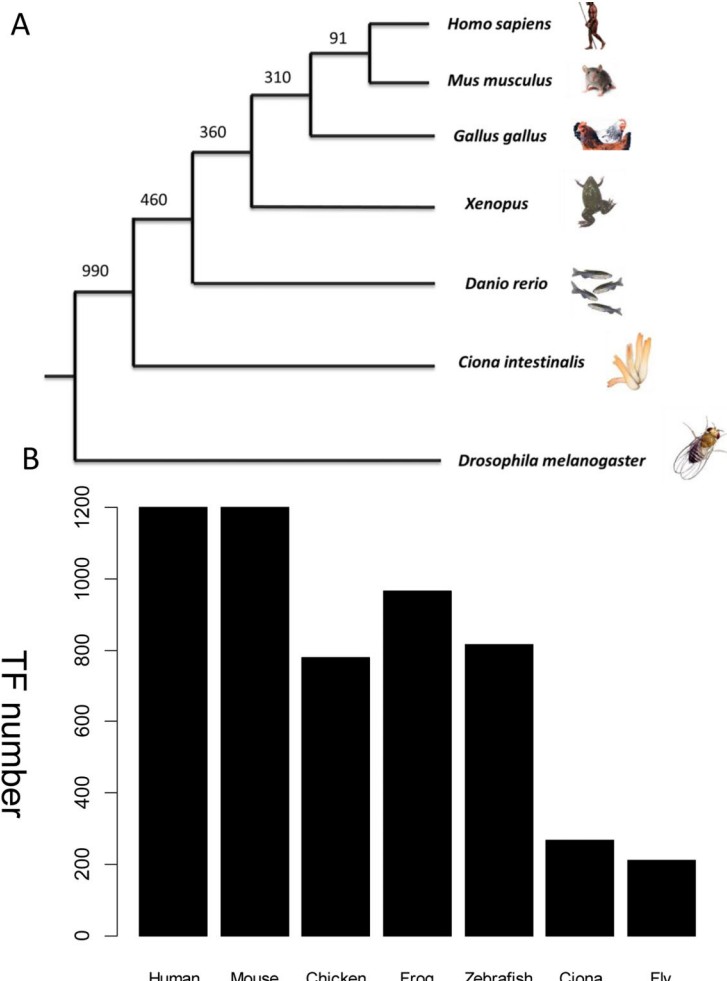

**Figure 3 TF distribution across species in the database.** (A) Phylogenetic tree showing the main animal models commonly used in heart development research and their evolutionary relationship. The divergence times in millions of years ago (Mya) are shown on the basis of multigene and multiprotein studies. Branch lengths are not proportional to time (B) Distribution of TFs across six species. All TFs have homologs in humans. The unit of Y-axis is TF number.

Search results directly link to the original database through the PWM raw database key. The CardioTFmatrix class contains 904 records, and these PWM files can be recognized by our local CardioSignalScan program to search for the motifs in genomic regions.

## Core cardiovascular TFs

The 1,200 mouse TFs were included in the cardiac TF dataset as the entry point to initiate deep annotation. To define a core set of cardiac TFs, we intersected four independent sources of cardiovascular TF collections. Inclusion of the resulting 81 TFs is supported by their expression status, phenotype annotation, expert recommendation and PubMed relevance (Table S1). We also performed DAVID functional analysis, and found that these TFs are particularly enriched in cardiac muscle differentiation (Fig. S2). 10–20% of these
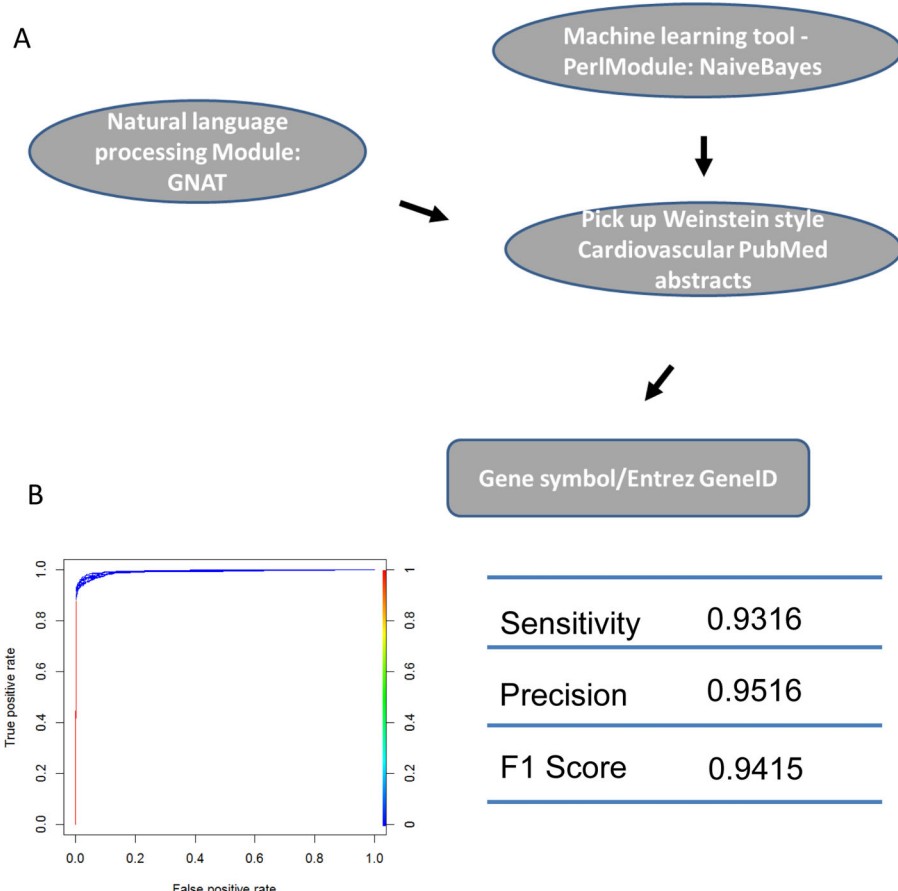

**Figure 4 Machine learning protocol used to select TFs described in Weinstein-like papers.** (A) The pipeline used to select TF gene symbols from Weinstein PubMed abstracts. First, a Naïve-Bayes module was used to select Weinstein-like papers from PubMed abstracts. Second, GNAT, a software that recognizes gene symbols, was used to identify all TF names from these Weinstein-like papers. (B) ROC curve and prediction performance judged by sensitivity, precision and F1 score.

TFs which are enriched in the Annotation Cluster 7 including Gata4, Gata6, Smad7, Nkx2-5, Tbx2, Tbx5, Foxc1, Foxp1, Prox1, Rara, Rarb, Rxra, Rxrb, Zfpm2. These 14 TFs, are annotated in the DAVID as being involved in cardiac muscle formation. As we know, the heart system includes the endocardium which is a specialized layer derived from endothelial cells. Cluster 8 from our DAVID analysis includes genes expressed in endothelial cells such as Smad5, Smad7, Meis1, Nkx2-5, Tbx20, Epas1, Foxc1, Foxo1, Hey1, Hand2, Hif1a, Mef2c, Prrx1, Prox1, Srf, Nr2f2, Tcf21, Vezf1, Zfpm2. Is this set of genes the minimum requirement for cardiac development? Indeed, these four sources of supporting evidence indicate that these TFs genes play a key role in heart development. We wanted to determine if these TFs display specific expression patterns in heart development. A heatmap was generated using seven RNA-seq data sets, including samples from embryonic cells, mesoderm cells, cardiac progenitors, nascent cardiomyocytes and adult heart tissue. This heatmap did not reveal any specific patterns (Fig. S3). In adult tissues, these TFs did not exhibit enriched expression in the adult heart. In the case of TFs

which are never expressed at any stage of heart development, no specific expression pattern was revealed by the boxplot assay (Figs. S4 and S5).

## Cardiovascular enhancers collected in this database version

Few enhancers have already been verified by traditional biological experiments, for example, by using transgenic expression of isolated DNA fragments in vivo to analyze temporal-spatial patterns. Therefore, the ChIP-seq method provides a high-throughput approach to delineate enhancer regions at the genome scale. A standard protocol was used to identify genome-wide locations of transcription/chromatin factor binding sites or histone modification sites from ChIP-seq data (Fig. S6). The present database houses 511,893 enhancer records, covering different stages of heart development. Searching for a single enhancer also provides a user interface to scan the TFBSs. The binding matrices provided in the list come from the core cardiac TFs. The flat text output file contains the matrix key for the TFs in the database and sorts these hits in the increasing order.

## DISCUSSION

We identified 5,442 TFs from six species, and integrated 904 PWM files from three PWM databases. We also collected 511,893 peak fragments for further analysis. The on-the-fly search tool was implemented to match the core cardiac TFBSs in the specified enhancer sequence. Our database provides a framework where users can query homology information for various TFs across species and PWM information corresponding to TFs and enhancers from high-throughput ChIP-seq data. The curation for cardiac enhancers and TFs facilitate future efforts to construct transcription network.

The database now contains the six species which are model organisms for studying heart development (Fig. 3A). *Ciona* is an ideal model used to study the early specification step in the *Mesp*-lineage. Zebrafish is well suited to perform imaging analysis and for performing quantitative study. The chicken is well suited for lineage tracing ex vivo. The mouse provides a well-studied mammalian model and is most similar to humans. The fly is a valuable model for testing new concepts and is easy to study at the genomic level. New models may be added in the future if they have unique advantage over the other models.

Previously, we constructed the CardioSignal database which collates cardiac factors driving genes expressed in a tissue-specific or quantitative manner. Most enhancers archived in the database are expressed specifically in the myocardium. CardioTF is a complementary database that accumulates cardiac TFs expressed in the epicardium, endocardium and myocardium. This information including PWMs can be used not only to find the features of enhancers in a machine learning approach (such the left-right patterning of the heart) but also to reconstruct regulatory network in systems biology.

We defined a core set of TFs using four independent sources, namely, RNA profiling, expert curation, PubMed abstract parsing and phenotype annotation to support their roles in cardiovascular development. In RNA-seq analysis, after pre-processing and post-processing of the adult heart data, the count table contained 48,995 genes. After filtering genes with FPKM > 1, there were 20,863 genes remaining. Roughly half of these annotated

genes were abandoned because their gene expression level was too low. To analyze PubMed abstracts, a Naïve-Bayes approach was used to classify Weinstein-style abstracts and then pick out the TFs embedded in those abstracts. Most journals containing the key words "heart" or "cardiac" or "cardiovascular" focus on heart pathology or physiology instead of development biology. For example, journals such as "Cardiovascular research," "Circulation," and "Circulation research," accept papers related to the adult cardiovascular system. Most papers published in these journals take a more translational approach which is oriented to bench-side work. Our analysis identified two heart related journals (which is obvious from their name) suggesting that the algorithm was successful in finding the wording pattern present in Weinstein abstracts. The results indicate that most developmental biology manuscripts relating to cardiovascular system are sent to specialized journals that focus on development. Top-tier cardiovascular journals are more likely to publish papers describing the adult cardiovascular system. Our text analysis, whether normalized by publication number or not, had a tendency of identifying journals favored by heart developmental biologists and journals that specialized in developmental biology (Table S1).

Traditional definitions of heart specific TFs can often be ambiguous and should not include TFs that only regulate cardiac muscle. In the present definition, heart specific TFs must be detected to be expressed in heart tissue which includes the endocardium and epicardium layers. Both DAVID Annotation Clusters 7 and 8 contain TFs involved in heart muscle or vascular formation. The DAVID analysis result did not reveal any other clusters with genes involved in kidney, liver or the formation of other organs. Even the expression of *GATA4* and *MEF2C*, which are de facto cardiac TFs, is not restricted to cardiomyocytes. These TFs are expressed in cardiac progenitors at a certain stage of development. The present approach is empirical and proposes a method which uses four independent data sources to identify true cardiac TFs based on their expression, phenotype, community opinion and PubMed abstracts. These 81 core TFs could be used further to support simulation study to infer the significance in future.

In general, the set of core cardiac TFs identified by these sources provide a roadmap for systems biology to construct a transcriptional network of heart development. Current approaches by the Sperling group or the Pu group only report three to four TFs based on ChIP-seq data (*He et al., 2011*; *Schlesinger et al., 2011*). Similar approaches by other genome biologists who tried to find cardiac enhancers on a genome scale have been reviewed elsewhere (*Wamstad et al., 2012*; *Wamstad et al., 2014*). However, the information generated from these studies is well below our knowledge of these core cardiovascular TFs, which have multiple sources supporting their role in cardiovascular development.

We archived 1,200 mouse TFs and wanted to determine at what stage of heart development they were expressed. Our preliminary analysis indicates that approximately 200 TFs have no evidence of their expression pattern, phenotype, expert recommendation, and PubMed abstracts. Whether these TF genes are expressed or play roles in heart disease requires further analysis.

The database still lacks cell lineage-based expression profiling data, which will quantify the expression level of various TFs and thus construct a 4-D dynamic expression pattern in vivo. This information could be combined with cell lineage-based ChIP-seq data to create a super-resolution of enhancer tomography.

## CONCLUSIONS

Modern translational medicine rests upon the progressive study of pathways and principles from model organisms such as yeast, fly, fish, and mice to clinical studies in humans. Therefore, we recruited TFs from six model organisms which are established models for research on cardiovascular development. The identification and collation of these well-annotated homologs from different organisms will enable investigators to better address the complexities of transcriptional network on heart development (*Wamstad et al., 2014*).

We hope that in the near future, single-cell sequencing data may provide comprehensive gene expression information with detailed temporal-spatial resolution, thereby providing insight into the transcription networks that contribute to heart development or heart diseases. CardioTF database try to collate these *cis* and *trans* information and take the initial steps in the construction of a comprehensive transcriptional network.

## AVAILABILITY AND REQUIREMENTS

CardioTF database is freely available on the web at http://www.cardiosignal.org/database/cardiotf.html.

## LIST OF ABBREVIATIONS

| | |
|---|---|
| **TF** | Transcription factors |
| **PWM** | Position weight matrix |
| **UML** | Unified modeling language |

## ACKNOWLEDGEMENTS

The author is grateful to Prof. Rutai Hui, Prof. Weinian Shou, Dr. Tingting Li and Dr. Jianxin Chen for their helpful comments. Special thanks to Prof. Paul Krieg (University of Arizona), Matija Brozovic (ANISEED), Prof. Carl J. Schmidt (BirdBase) and Zichao Sang (CardioSignal) for their comments or suggestions on data curation. I would also like to thank Dr. Rosannah C. Cameron at the Albert Einstein College of Medicine for her assistance editing the manuscript.

### Funding

This work was supported by the National Natural Science Foundation of China (Grant number 31000644 to Y.Z.). The funders had no role in study design, data collection and analysis, decision to publish, or preparation of the manuscript.

## Grant Disclosures

The following grant information was disclosed by the authors:
National Natural Science Foundation of China: 31000644 to Y.Z.

## Competing Interests

The author declares that he has no competing interests.

## Author Contributions

- Yisong Zhen conceived and designed the experiments, performed the experiments, analyzed the data, contributed reagents/materials/analysis tools, wrote the paper, prepared figures and/or tables, reviewed drafts of the paper.

## Data Deposition

The raw data has been supplied as Supplementary Dataset Files and can also be downloaded from: http://www.cardiosignal.org/download/download.html.

## Supplemental Information

Supplemental information for this article can be found online at http://dx.doi.org/10.7717/peerj.2339#supplemental-information.

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
