# Peer review of "CardioTF, a database of deconstructing transcriptional circuits in the heart system"

_PeerJ, doi:10.7717/peerj.2339_

## Round 0.1 · original submission · Major Revisions

As you will see all three reviewers have found major problems with the manuscript as it stands. Please take care to address all their comments before re-submission.

Staff Note: Given the comments about the quality of the English language, we also suggest that you have a native English speaker review, or an Editing Service, look at the text

Reviewer 1 ·

Basic reporting

Overall, the article is poorly written, containing numerous spelling mistakes, grammatical errors and structure inappropriate for a research article. There are a number of references that have not been cited throughout. The article would benefit from further review to make it more accessible.

The introduction does not clearly articulate the need to centralise these data and nor are the results clear.

Experimental design

Most concerning is the lack of appropriately described methods and many missing references. Currently there is no way that another individual could reproduce the results given the methods description at hand.

Validity of the findings

The author largely describes CardioTF, a web based database containing information that the author claims to be specific to cardiac transcription factors and transcriptional output to enable systems biology research. The data is based on information collected from literature. The author argues that collection of these data is required to construct transcriptional networks. Although it is agreed that such information being located in a central location is of value, there are many fundamental questions throughout the article which require addressing, most of these relate to the very poor level of detail surrounding the methods and results. Thereby it is unclear what the actual results are. At present there are two distinct parts to the manuscript, 1) being the collection of data in the database and 2) being the text based classification of abstracts. The research component surrounds the classification of journals based on abstract submission to the Weinstein conference. Neither of these two components have been sufficiently explored. A very poor discussion which does not include any discussion surrounding the findings doesn't fulfil the requirements as a research article and requires major revisions to be considered. The article would therefore benefit from elaboration to what the actual results are, rather than what is largely a description of the data collected - which is weak as it currently stands.

In addition, the following comments are provided for the author to facilitate revision:

Lines 27-28. How are TF to PWM annotations and the ambiguities of correctly associating these dealt with?

Lines 26-27. “Mouse homologs were singled out for defining the core TFs by intersecting four independent data sources”. This is in the abstract and either makes no sense or is poorly explained.

Lines 39. Adult cardiac diseases cannot always be traced to variants. Indeed this is an active area of research and the sentence needs to be toned down to reflect this.

Lines 42-44. Expert annotation. What is this and how was it performed? Again this is an example of poorly described methods.

Line 44. I assume the author means the field has experienced growth, not cardiac development.

Line 48. Assume this is “To date”, not “To data”.

Line 51. This is not correct. There have been other attempts that need to be acknowledged here: http://www.ncbi.nlm.nih.gov/pubmed/25369032

Line 58. The author refers to a simple graphical output. In reality there is no valuable information in the display. When testing the website, there was no graphical output.

Line 63. This makes no sense. It is obvious that cardiac TFs would play a role in cardiac development. What is the point here? Where were the cardiac TFs previously defined?

Line 68. Refers to a reference for MESP1, but it is unclear what the intention of this sentence is with regards to retrieval of cardiac TFs.

Lin 79-80. Use of Weinstein abstracts. 954 abstracts. What were they? How did they check them?

Line 83-84. Defines how the negative set was constructed which required manual annotation, however there was no mention of what the journals were or criteria for manual annotation. These negative control journals, the periods of time they were selected and preferably the actual references to each journal article/abstract used should be provided to enable reproducibility of results.

Line 88. Dietterich approach for model validation. Where are the results for this?

Line 89. Word frequency was calculated. Was this all words or only specific words? The consequences of either approach has not been discussed.

Line 90. A separate test set was used? Is the 20% test set or another test set? This is not clear at all what was done and why.

Line 98. How many genes were greater than 1 FPKM. Why was this threshold chosen?

Line 103. What is the in-house cardiosignalscan program? How does it work?

Line 107. What algorithm, how does it work and where it is located?

Line 117. When were Bowtie or Bowtie2 used? Similarly mm10 and hg10 need to be defined better. Was this using repeat masked or non-repeat masked, all chromosomes? This another example of poorly describes methods.

Line 120. What version of MACS was used? What probabilities if any were used to determine significance of peaks detected?

Line 129. Reference is missing for UML. Is the description of UML here really a result or a method/material?

Line 134. Modelling principles references are missing.

Line 141-149. How was the search engine created? Does the author mean that query forms were created? Should this section be in the methods, what is the result here?

Line 151. Reference missed and no explanation provided, what is the Wingender’s annotation set?

Line 152. How many TFs were excluded by this criteria? Was there sound evidence in their exclusion?

Line 156. There is no methods description accompanying Figure 3A and nor is it discussed in text.

Line 158. References missing.

Line 163. How? Were these based on highest predictive performance? How can you determine that it is classifying cardiac correctly versus other discriminators in the abstracts used for training the models that are specific to Weinstein-style papers? There is no curated gold standard to assess this, i.e. there is no real validation. What happens when the bayes model is re-implemented based on just the cardiac TF names searched for in the papers?

Like 164. It is assumed that the machine learning method is Bayesian?

Line 166. Only two from 30, what does this indicate? Furthermore in Table S1, it is unclear what the statistic represents here? Is this the number of abstracts classified as Weinstein like? If so, is this biased towards journal publication rate. If this is the case then the statistics need to be normalised.

Line 177. How does the program find the PWM matches? How do you deal with redundancy of PWMs from different sources or matching different but related TFs?

Line 181. What four independent sources?

Line 182-183. Evidence in Table S1 – there is no expression evidence in this table and it is unclear what the author is referring to in the supplementary table.

Line 185. Figure S2 doesn’t show complete data and isn’t the highest ranked cluster. Comment on this and include other relevant data. Furthermore, only 10-20% of the TFs might have been captured in any term here. What were they? This data should be provided as supplementary tables.

Line 189-192. These data suggest that the selection of TFs using the method did not identify heart specific TFs. Explain the implications of this further. How does this distribution compare to randomly selected TFs for example?

Line 196. Reference is missing.

Line 197-200. Weak description of the methods that is not supplemented by figure S6. To be an adequate result, the description of the enhancer records needs to be significantly expanded.

Line 202-204. Where did these numbers suddenly come from? This should be in the results section.

Line 216. Many references and efforts by other have not been reported here.

Figure S1. What do the different lines represent?
Fig 2. The enhancer regions is of no values here. Are these from different species, what do the E000XXX numbers represent and what are the axis.
Fig 4. The pipeline is not a very good representation. There are multiple line in the ROC curve, how was sensitivity/specificity/F1 score calculated? Are these averages?

·

Basic reporting

There are a number of spelling and grammar issues throughout this manuscript.

The manuscript contains some relevant background information but also many unsubstantiated and confusing statements such as:

"In addition, cardiac development has experienced accelerated growth that can
45 be attributed to the use of various animal models."

This statement is made without any reference to substantiating literature.

"To date, a limited number of databases have been dedicated to collecting similar data aspects.For example, BloodChIP [7] is a database of comparative genome-wide TF binding profiles in human blood cells. CistromeMap [8] is a knowledgebase and web server for ChIP-Seq and DNase-Seq studies in mice and humans."

In my opinion, large-scale studies such as ENCODE and modENCODE which also contain comprehensive online tools should be acknowledged here.

There are also a number misleading statements that appear to ignore the literature entirely such as:

"However, to our present understanding, few efforts have been committed to build a database which collects transcriptional information across species, thereby limiting the benefits from evolutionary perspectives."

There are a very large number of such resources which collate both TFBS and gene expression across species and have also several evolutionary studies (eg Babu et al 2003, Chen et al 2007)

Experimental design

The manuscript contains several parts of the design which are poorly explained for instance (but not limited to):

"We also documented the mouse TF expression status from four independent sources at different developmental stages"

But nowhere in the manuscript are these developmental stages introduced or explained.

"which were manually selected from PMC Open Access Subset at NCBI. The negative group includes 57080 abstracts."

No description of the "manual selection" is provided

"A separated test set using optimized parameter was used to assess the algorithm performance."

No description of the optimisation is given

Wingender’s annotation set was used as a benchmark to recruit TFs across species. Human-specific TFs, defined as those with no orthologs in the mouse genome, were discarded because no model system could be used to verify their function in vivo. Therefore, 1200 mouse TFs were

No citation for this annotation set is given of the optimisation is given.

"Enhancer regions were defined as the ChIP-seq signals."

This description of enhancers is not one that matches with the current understanding and seems to demonstrate a lack of understanding in this area. In the title the author refers to Histone-ChiP but this is not followed up in the text

"NCBI has its own collection of all ortholog gene collections using its unpublished algorithm. TFs from mouse, human and zebrafish are annotated by NCBI [20]. "

This reference is not related to this statement. Also, the details of the orthologs sets from NCBI are no unpublished and the author should familiarise themselves with this work before making a statement such as this.

"Few enhancers have already been verified by traditional biological experiments, for example, by using transgenic expression of isolated DNA fragment in vivo to analyze temporal-spatial patterns."

Again this statement is not substantiated and incorrect.

Validity of the findings

Given that the descriptions of the methods is lacking it makes judgeing the validity of the findings very difficult. For example, when considering the ability of this approach to discover "Weinstein-like papers," the description of both the datasets and the tests performed are not adequate to judge the results.

The description of enhancers seems inadequate and as such it seems difficult to assess whether the inclusions of these "enhancer" regions are really beneficial to the community.

The authors claim to have identified 81 core TFs that:

"their expression pattern, phenotype annotation, expert recommendation and PubMed relevance (Table S1)"

but then go onto to conclude that these TFs:

"did not reveal any specific [expression] pattern" , "did not exhibit enriched expression in the adult heart" and even that some core transcription factors are "are never expressed at any stage of heart development"

With no explanation as to why these TF remain as core regulators despite these observations.

With respect to the online tool, I found it very difficult to understand the relevance of each of the pages. For instance, http://www.cardiosignal.org/cgi-bin/cardiotf.pl?page_id=2&db_text=176843 contains no information about what is being displayed which makes the page very difficult to use.

Additional comments

The goal of this paper is worthy of publication, but in my opinion, the execution is far from adequate. The paper contains many unsubstantiated and incorrect statements that make judging the quality of the rest of the work very difficult. The tool also requires a great deal of refining in order to make it understandable and usable by the community.
The manuscript would benefit from a much clearer description of the goals of the database, the available tools that it provides to the community and an example of how these tools can be used.

Reviewer 3 ·

Basic reporting

- The article requires extensive review at the language level as several sentences are not written in clear English and include typos. Below are a few examples:
o “high-through-put” instead of “high-throughput”
o “To data” instead of “to-date”
o “resolution of enhancer tomography”, I assume the author meant “topology”?
o “The nature language” instead of “natural language”
o “matrces” instead of “matrices”
o The title needs revision
o Ciona not in italics

- Publications from International consortia such as ENCODE, FANTOM and Epigenomics Roadmap were not cited although they include a wide-range of genome-wide assays pertaining to heart development. Similarly, sophisticated genome-browsers such as EnsEMBL and UCSC are not cited either, although they provide a customable search interface to browse for the result delivered by the consortia mentioned above to generate a cardiac-specific output.

- Several “custom-made” perl scripts have been mentioned throughout the manuscript but they were not provided or made available through a resource (e.g.: github) as requested by PeerJ’s data sharing policy.

Experimental design

The author describes a pipeline aiming to annotate cardiac transcription factors and stores the output in a database repository, which is made accessible through a web interface that provides information on the transcription factor, its binding site and genome-wide target sequence. There is indeed a lack of resources collecting information relating specifically to cardiac transcription factors.
Therefore the author’s effort to fill that gap is commendable; in particular, the use of text mining and machine learning to annotate cardiacTFs is sound and adds value to the current knowledge in that area. The methods pertaining to this specific aspect of the study are described in depth, and database schema is made available as a mySQL dump.
However, the methods used to analyse other datasets are poorly described. Namely, R and Mas5 were mentioned as methods to process the microarray data, although these are generic software will different options and no citations were provided to describe the analysis performed on these arrays. Most strikingly, regarding the ChIP-seq experiments, it is not clear which datasets have been used in this study: TF-ChIP and histone ChIP are mentioned in the paragraph titles but no references are provided for any of these datasets.

Validity of the findings

This study describes a database harboring information on cross-species cardiac transcription factors, binding sites and target sequences.
The content of the database is composed of a core-list of 81 cardiac transcription factors and their orthologous counterparts which were determined by the well-documented methods in the Materials and Methods section, and in the supplementary table 1. Binding sites were provided by simple links to PWM databases. However, there is no information on how the target sequences (enhancers) were associated these core TFs. More specifically, how did the author associated cardio vascular enhancers obtained from histone marks for specific cardiac TFs? The lack of information for this aspect of the work is a major concern regarding how the database has been populated for enhancer information.
As a repository, the database relies on the web-based query interface for the data to be accessible. However, the interface is cumbersome and does not allow for searches that permit the cross-species interrogation as claimed by the author. For example, no wildcard search is available to browse for different nomenclatures for the same gene in different species. In addition, links to PWM databases do not specifically target the matrix but lead to the whole PWM repository, therefore researchers might as well perform the search directly in the PWM repository. Finally, the graphical output to visualize the enhancers of the genes of interest is obscure. There are no legends describing what do the horizontal bars refer to.

Additional comments

In summary, this work is a mixed of sound analysis (e.g: machine learning section) and very poorly described methods (e.g.: enhancer section). Hence it requires major revisions, from the writing of the manuscript itself (language, typo, references), to an extensive description of all the methods so they can support the conclusions made (i.e: the contents of the database), and finally it needs a drastic improvement to the web interface.

---

## Round 0.2 · Major Revisions

The paper has now been seen by all 3 reviewers and all have requested major revisions. Please make sure to address all of the requests this time. Reviewer 1 has given extensive instructions on how to improve the manuscript please try to address all of these points and take care to ensure the changes are reflected in the manuscript and not only in the rebuttal. The reviewers should not be acknowledged by name in the acknowledgements, please remove this.

Reviewer 1 ·

Basic reporting

With regards to language/grammar, it is evident that the earlier parts of the manuscript have benefited from some changes, there are a number of areas that require further improvement, especially evident in the latter sections (e.g. line 266).

Furthermore, although comments in the rebuttal address changes, many changes have not translated into the manuscript. One example, a previous comment “Lines 39. Adult cardiac diseases cannot always be traced to variants. Indeed this is an active area of research and the sentence needs to be toned down to reflect this.” Which was responded with “I have re-written the sentence as: “Adult cardiac diseases might be traced to variants”.” has not been changed in the actual manuscript.

Please pay careful attention to this upon your next submission. The methods, results and discussion need to be made clearer.

Experimental design

The authors response, and my response, labelled "RESPONSE", to their actions are below:

I have uploaded all the Perl scripts and R codes to GitHub (https://github.com/zhenyisong/). I also have uploaded the raw data, including Weinstein meeting abstracts (positive_test_data plus positive_training_data), and negative dataset (negative_test_data plus negative_training_data), in zip format and also intermediary files which contain cross-validation result as well as other public data to the CardioTF database server (http://www.cardiosignal.org/download/download.html). The raw data and source code can be used to verify my findings. I also revised the methods section to include further details about the protocols which will help reader to understand the data processing steps and should make the results reproducible.

RESPONSE:
This is a good start. However the Github link is not referenced in the paper and should be included. Furthermore, there are multiple instances where the response has been given here but not included in the article – see below instances.

This sentence should be considered in the context of my previous work on the cardiosignal database [1]. Even in cardiovascular community, this concept is a little bit confusing. In cardiosignal database, we aimed to collate the cardiac specific enhancers which drive gene expression in cardiomyocytes. At that time, we defined cardiac specific transcriptional factor as genes that directed expression of genes in the myocardium. As we know, the heart includes myocardium, epicardium and endocardium. These three layers in early stage constitutes of full structure of the heart. At least four heart-specific lineages have been characterized, including cardiomyocyte, endothelial cell, epicardial cell, and fibroblast cell, the latter of which is derived from epicardial cells through EMT. In our cardioTF database, I collated transcription factors which are expressed in all layers of heart and not limited to cardiomyocytes. The goal of the CardioSignal database is to use a machine learning approach to find cardiac enhancers at the genome scale. In contrast, the cardioTF database is constructed to study systems biology of transcription regulation. The cross-talk between different layers will also be explored using this platform. The CardioSignal database is currently under repair, and the previously defined cardiac TFs can be retrieved from the MySQL dump (frozen at 2006-07-01) at the download page of the CardioTF server.
[1] Zhen Y, Wang Y, Zhang W, Zhou C, Hui R. CardioSignal: a database of transcriptional regulation in cardiac development and hypertrophy. Int J Cardiol.2007 Apr 4;116(3):338-47.

RESPONSE: Yes, there was a reference lacking. Unfortunately the reference, its meaning and source has not been included in the updated manuscript and should be included for clarity.

These abstracts were extracted from the meeting abstract book from 2010 to 2013 by hand. The raw data including 954 abstracts was uploaded to CardioTF server (http://www.cardiosignal.org/download/download.html). I indexed all the abstracts and randomly assigned them into a test and training (cross-validation) set. The Weinstein meeting focuses on cardiovascular development and abstracts are organized by research theme. I therefore was able to quickly identify abstracts that were about heart development.

RESPONSE: This needs to be stated in the methods not in the rebuttal: “abstracts were extracted from the meeting abstract book from 2010 to 2013 by hand”

I only used one test set (20% test set). This is a mis-expression and led to misunderstand my true meaning. The test set was used to calculate the sensitivity and specificity of the model. I did not use the cross-validation set (80%) to assess the final performance of the model. A separate, independent set from the cross-validation set, here referred to the test set was used to assess the performance of the naïve-bayes model.

RESPONSE: In that case, the following would be clearer as a suggestion: “The withheld test set was then used to test the algorithms performance”.

In adult heart, after pre-processing and post-processing, RNA-seq data contains 48995 genes. After filtering with FPKM >1, there are 20863 genes left. Roughly, half of these genes were abandoned because their expression level was too low. In Biostar, one old post (https://www.biostars.org/p/61192/) said that most of the papers arbitrarily define expression threshold i.e, >1 FPKM/RPKM to identify an expressed transcripts. I used this rule in my study.

RESPONSE: This is a result then. Also the method for defining the threshold is arbitrarily set and should be stated in the methods.

This cardiosignalscan program was released in my previous publication [1] as the assistant tool to recognize the matched motifs between PWM files. Because the CardioSignal database is now under repair, this program is currently inaccessible. Actually, the program is a brute force approach to detect motifs embedded in DNA, and the running time of the program is increased exponentially. Thus, it is unsuitable to use it to scan DNA sequence longer than 3000bp. I uploaded similar script using the same algorithm as the cardiosignalscan to GitHub (cardiophylo.pl). In my recent work I used MOODS (Motif Occurrence Detection Suite) which claims to be increased linearly according to input sequence length. This program is developed by Dr. Janne H. Korhonen [2].
[1] Zhen Y, Wang Y, Zhang W, Zhou C, Hui R. CardioSignal: a database of transcriptional regulation in cardiac development and hypertrophy. Int J Cardiol.2007 Apr 4;116(3):338-47.
[2] Korhonen J, Martinmäki P, Pizzi C, Rastas P, Ukkonen E. MOODS: fast search for position weight matrix matches in DNA sequences. Bioinformatics. 2009 Dec 1;25(23):3181-2.

RESPONSE: The author needs to reference the papers in addition to the brief description that is now in the methods.

Line 107. What algorithm, how does it work and where it is located?
This algorithm of Homologene is implemented and maintained by NCBI staff. I did not install the program locally. Instead, I only retrieved its comparison result to get homologs for each species. It was stated that the algorithm was not published. Here is the original description for the algorithm [1]: “Homologene [6] uses a pairwise gene comparison approach combined with a guide tree and gene neighborhood conservation to group orthologs, but the details of their methodology are unpublished.”
[1] Altenhoff AM, Dessimoz C. Phylogenetic and functional assessment of orthologs inference projects and methods. PLoS Comput Biol. 2009 Jan;5(1):e1000262.

RESPONSE: The point is that any reference to the method is missing. Add a link to homologene and state NCBI Homologene in the text.

I used pre-processing software (FastQC) to estimate the read length in raw data. If read length is above 50bp, Bowtie2 was used. Otherwise, Bowtie was used. mm10 and hg10 are genome build versions used by UCSC. Index and annotation files for Bowtie2/Bowtie were downloaded from Illumina iGenomes project. Genome sequences from UCSC are repeat-masked with lower-case characters.

RESPONSE: This needs to be stated in the methods, not just here.

MACS1.4.2 was used to process all the ChIP-seq data. The default cutoff for the p-value is 1e-05. The default value was used in all ChIP-seq analysis. I used the protocol mainly adapted from this reference [1].
[1] Feng J, Liu T, Qin B, Zhang Y, Liu XS. Identifying ChIP-seq enrichment using MACS. Nat Protoc. 2012 Sep;7(9):1728-40.

RESPONSE: Again, this needs to be stated in the methods, not just here.

Wingender’s annotation set can be accessed by visiting the following web site (http://tfclass.bioinf.med.uni-goettingen.de/tfclass). I curated the 1564 TFs from this dataset, hence it was named Wingender’s annotation set. This dataset is comprehensive in annotating human TFs. The version I used as the raw data was uploaded to the CardioTF server. I have added the reference.

RESPONSE: Reference needs to be updated – refers to wrong reference.

The frozen version of Wingender’s annotation set contained 1564 human TFs. However, only 1513 TFs have corresponding Entrez gene records and 51 TFs have no Entrez gene records. I then used homolog annotation from NCBI (HomoloGene data) and excluded 313 human TFs which have no counterpart in mouse, thus leaving 1200 human TFs (1513 – 313 = 1200) which have mouse homologs.

RESPONSE: State the numbers in the text, not just here.

This picture (Figure 3A) is generated manually according to this article [1]. Branch lengths are not proportional to time. Figure 3A represents the animal models used by cardiovascular community. The models were discussed by Drs. Schulz and Black in Seminars in Cell & Developmental Biology (Volume 18, Issue 1, February 2007, pages 1–2).
[1] Hedges SB. The origin and evolution of model organisms. Nat Rev Genet. 2002
Nov;3(11):838-49.

RESPONSE: Figure 3A is still not discussed, it should be removed if not discussed.

I have added the references accordingly.

RESPONSE: Reference numbering seems to be off by one.

Yes. It is naïve bayes. I used the Perl Module from CPAN (Algorithm::NaiveBayes).

RESPONSE: State “naïve bayes” in the text.

In my previous paper [1], I used Cardiosignalscan to find the PWM matches. This program enumerates all positions in both strands of the string to find the PWM matches. The source code is available from GitHub (cardiophylo.pl). We list all the PWMs from those three sources for each TF. We currently do not use the program proposed by Zhang lab [2] to check the similarity of PWMs and reduce the redundancy in the collection of PWM files.
[1] Zhen Y, Wang Y, Zhang W, Zhou C, Hui R. CardioSignal: a database of transcriptional regulation in cardiac development and hypertrophy. Int J Cardiol. 2007 Apr 4;116(3):338-47.
[2] Schones DE, Sumazin P, Zhang MQ. Similarity of position frequency matrices for transcription factor binding sites. Bioinformatics. 2005 Feb 1;21(3):307-13.

RESPONSE: This relates to the earlier comment of citing the article.

In this case I was referring to gene expression status and Table S1 does not provide quantitative values of gene expression. To generate the gene expression status I combined the microarray data and RNA-seq data. For microarray data, a MAS5 call was calculated to determine whether the gene is present or not. In RNA-seq data, FPKM < 1 was set as the cutoff value to judge whether the gene was expressed or not.

RESPONSE: State after expression pattern (data not shown).

In responding to Review#1’s question, I re-ran the DAVID analysis (version 6.7) using the 81 TFs as the input gene list, official gene symbol as the identifier and entire mouse genes set as the background. I choose functional annotation clustering to export the results. The classification stringency was set to default (medium). There were minor changes in the order of the clusters. However, the top ranked clusters were still those containing terms such as transcription or positive/negative gene regulation, thus I omitted these clusters in my old manuscript. In the updated version, I revised Figure S2 accordingly; now it is cluster 6 and 7. To reveal more details and provide an unbiased explanation of the result (DAVID analysis of the 81 core TFs), I have placed the whole report in the supplemental data (Supplemental_Document.txt). Those 10-20% of the TFs which are present in the Annotation Cluster 7 include Gata4 ,Gata6, Smad7, Nkx2-5,Tbx2 ,Tbx5,Foxc1,Foxp1,Prox1,Rara ,Rarb,Rxra,Rxrb,Zfpm2. Those 14 TFs, as annotated by David database curators, are involved in cardiac muscle formation. As we know, heart system at least include endocardium which is specialized layer derived from endothelial cells. This argument is supported by presence of cluster 8 from my David analysis which includes genes expressed in endothelial cells such as Smad5,Smad7,Meis1,Nkx2-5,Tbx20,Epas1,Foxc1,Foxo1,Hey1,Hand2,Hif1a,Mef2c,Prrx1,Prox1,Srf,Nr2f2,Tcf21,Vezf1,Zfpm2. I have also added Supplemental_Figure_S7.pdf (Clusters 1-3), Supplemental_Figure_S8.pdf (Clusters 4-6) and Supplemental_Figure_S9.pdf (Clusters 9-13).

RESPONSE: Methods described here need to be included under a heading of Gene Ontology Analysis in the Methods section.

To generate enhancer records, I first search the PubMed and GEO database to find the research articles and the corresponding raw ChIP-seq data. I used FastQC to perform pre-processing. After that, Bowtie/Bowtie2 was used to align the reads to the genome. If read length was above 50bp, then Bowtie2 was used.
Bowtie method was following:
bowtie -m 1 -S –q mm10 -p 4
The script was written to submit the job to the supercomputer. After that, the format transformation was performed:
samtools view -bS -o tbx20_positive.bam positive_tbx20.sam
When possible, the control files were merged:
samtools merge out.bam in_1.bam in_2.bam in_3.bam
After completing the task of MASC program, annotatePeaks.pl in HOMER was used to parse the bed file from the MACS output. Then the parsed results were dumped into the MySQL table.

RESPONSE: As per above, include/make clearer in the methods.

These numbers are statistics from the CardioTF database. They are also mentioned in results section. According to the writing style of many research articles, most researchers summarize their main findings in the same order in the results and discussion sections. As my studies focus on database curation, I have mentioned my statistics in both methods and results sections.

RESPONSE: This is the first time “5442” is mentioned.

I used 5 ⨉ 2 cross-validation approach to select the parameter. There should be ten curves in Figure S1B.

RESPONSE: This needs to be stated in the figure legend.

This is the output for users to retrieve the corresponding enhancer sequences. They are from the same specie, but might be from different experiments. A user can check the experiment information by clicking the E0000XXX link which represent the primary key (ID) for this enhancer region, thus allowing the user to save it and retrieve the information later.

RESPONSE: Again this should be stated in the figure legend.

The sensitivity /specify /F1 score was calculated based upon the test set (20%). So the result does not represent the averages across 5 ⨉ 2 cross-validation.

RESPONSE: If this is based upon the single test set, then why does it appear that multiple lines are present in Figure 4b?

Validity of the findings

Mesp1 is the indicator of pre-cardiac mesoderm, which, to our knowledge, is the earliest marker that labels all cardiac lineages. [1]. In Ciona, the Mesp enhancer specifies heart precursor cells. In my CardioTF database, we aim to collect all the information derived from Mesp lineage. This definition is rigorous and not ambiguous in the context.
[1] Saga Y, Kitajima S, Miyagawa-Tomita S. Mesp1 expression is the earliest sign of cardiovascular development. Trends Cardiovasc Med. 2000 Nov;10(8):345-52.
[2] Satou Y, Imai KS, Satoh N. The ascidian Mesp gene specifies heart precursor cells. Development. 2004 Jun;131(11):2533-41.

RESPONSE:
Beyond Mesp’s role, this is still not clear to the use of the Mesp enhancer. Either the reference to Mesp is in the wrong place or Mesp was not used in the definition of the cardiac TFs. If it is the latter, it is missing from the methods.

To my knowledge, most journals containing the key words “heart” or “cardiac” or “cardiovascular” focus on heart pathology or physiology instead of development biology. For example, journals such as “Cardiovascular research”, “Circulation”, and “Circulation research”, accept papers related to the adult cardiovascular system. Most papers in these journals take a more translational approach which is oriented to bench-side work. Identification of two heart related journals (which is obvious from their name) suggests that the algorithm was successfully in finding the wording pattern present in Weinstein abstracts. The results indicate that most developmental biology manuscripts relating to cardiovascular system are sent to specialized journals who focus on development. Top-tier cardiovascular journals are more likely to publish papers describing the adult cardiovascular system. The statistics in Table S1 represent the number of abstracts classified as Weinstein like. In my updated Table S1, I normalized the data using the number of annual publications multiplied by a scale factor (1000). Again, among top 30 journals, there are two heart journals (Eur J Echocardiogr, and Heart Rhythm). Therefore the above conclusion still holds true. Developmental biologists prefer to sending their manuscripts to development-specific journals.

RESPONSE: This should be discussed in the discussion.

In the text analysis, I tried to curate all the cardiac transcription factors from the PubMed abstract. I assume that if an abstract from PubMed is classified as Weinstein-like abstract, this paper will contain information about heart development while not about other things, namely, liver or kidney development. If the abstract is classified as Weinstein-style, then the gene names in the abstract are much more likely to be involved in cardiac development. After cross-validation, I chose a parameter based on average predictive performance. The parameter was not determined by the highest predictive performance of a single training round. This parameter resulted in ACC (Accuracy) equal to 0.99. In this sense, the implemented algorithm together with GNAT tool was used to find cardiac TF names in abstracts. I assumed that gene names mentioned in the Weinstein abstracts were all cardiac genes. Reviewer#1 suggested using cardiac TF names to search the cardiovascular papers. Reversing the search to find articles regarding cardiovascular development is a good idea. However, only using cardiac TF names in this context will definitely lead to false positive results. The reason is that currently there are no cardiac TFs which are specifically expressed in the heart and only studied by cardiac biologists. If grouping them (compound search), most search result will identify the research articles and research groups regarding cardiac TFs. Those research groups prefer to study cardiac transcriptional factors, namely, Eric Olson (Mef2c, Hpox) or William Pu (Gata4,Wt1).

RESPONSE: Discuss implications in the discussion

Line 189-192. These data suggest that the selection of TFs using the method did not identify heart specific TFs. Explain the implications of this further. How does this distribution compare to randomly selected TFs for example?
In my opinion, the definition of heart specific TFs can often be ambiguous and should not contain TFs that only regulate cardiac muscle. In my rigorous definition, heart specific TFs should play specific role in establishing Mesp-1 lineage, this includes regulators of both the endocardium and epicardium layers. Both David Annotation Cluster 7 and Cluster 8 contain TFs involved in heart muscle or vascular formation. The David analysis result did not reveal any other cluster with genes involved in kidney or liver or formation of other organs. Even the expression of GATA4 and MEF2C, which are de facto cardiac TFs, is not restricted to cardiomyocytes. These TFs are expressed in cardiac progenitors at a certain stage of development. Reviewer#1 suggests doing the enrichment analysis by inferring the p-value using a simulation study. Actually, we do not have true cardiac TF set which is accepted as the benchmark by the research community. My approach is empirical and proposes a method which uses four independent data sources to identify true cardiac TFs based on their expression, phenotype, community opinion and PubMed abstracts.

RESPONSE: This results needs to be discussed in the discussion.

Additional comments

Although it appears that many of the minor changes have been corrected, please do not address items in the rebuttal without translating those into the manuscript.

·

Basic reporting

(1) There are still a couple of spelling errors in the manuscript but this is vastly improved from the first submission (some examples below)
To address how information will be stored nad how the lements will related to one another, we used the unified modeling language (UML) to describe the high-level database
Thus, the database is able to perform the key fucntions required to construct a transcriptional network of heart development.

(2) I am unable to find any figure legends.

(3) I still feel that the descriptions of the methods is extremely vague in places, for example:

Enhancer regions were defined by ChIP-seq signals.

Here there is no description of which types of Chip-Seq signals are included and why. In the authors rebuttal letter it is stated that "In Bing Ren’s work [1], they defined the H3K4me1 or H3K27ac modifications as markers for enhancer elements. In William Pu‘s work [2], they assume that the cluster of SRF, GATA4 and MEf2C are the hallmark of cardiac enhancer. " which may be the case but it is not explained anywhere in the manuscript which Chip-Seq data is used. It is certainly not the case that a cluster of any chipseq data is a mark for an enhancer and from the manuscript it is impossible to tell if this is the assumption being made, even within the supplementary figures it is only ever refererred to as Chip-Seq with no further details.

(4) The additional details of the methods that have been described in the rebuttal letter are not always present in the manuscript. Whilst I appreciate that the author is trying to respond to concerns of the reviewers, these details also need to appear in the manuscript. For example:

In the rebuttal letter, the author says "MACS1.4.2 was used to process.." but this detail in not in the paper. This is only one example, I will not outline each of them, but the rebuttal letter contains many useful descriptions of the methods that need to be in the paper.

Experimental design

The manuscript clearly describes that its purpose to provide cardiovascular specific regulatory information. However, as I have outlined above I have still have concerns about the description of the methods. There are far too many general statements such as "various developemetal stages". Furthermore it is not correct to report "We also performed DAVID functional analysis, and found that TFs are particularly enriched in cardiac muscle differentiation" without the relavant details. These sorts of statements need to include details on (a) levels of significance (ie p-values) and (b) methodological details such as what background was used in order to valid. Without details such as these reading the paper, understanding clearly the process and making a judgement about the validaity of the results is very difficult.

Validity of the findings

Please see my comments above.

Additional comments

I do not think it is necessary/appropriate to acknowledge me in the manuscript directly, please remove this.

This work will be of use to the field, but it is important that all of the methods are clearly and fully described in order to allow a reader to fairly judge the results, at present I feel that the manuscript doesn't achieve this.

Reviewer 3 ·

Basic reporting

The English has improved however some sections are still unclear, for instance: line 390 what does "in the order" mean?

Experimental design

It is still unclear how enhancers have been assigned to TF, which is clearly a feature that is displayed using the following parameters:
"Cardiac gene enhancer"
"Cardiac gene name"
"Gata4"
The output displays "gata4 Enhancer search result" and lists a series of enhancers somewhat associated with gata4. How was this association derived? Are these enhancers that overlap with gata4 chromosomal location? If yes, upstream and downstream regions around the gene should also be displayed since enhancers are usually located in non-coding regions, very often in intergenic regions.

Validity of the findings

The manuscript is overall improved.

---

## Round 0.3 · Major Revisions

Please take special care to address to list up the datasets used in the paper.

Also please clarify about MESP1. If it has not been used then remove all references to it.

Reviewer 1 ·

Basic reporting

Overall the manuscript is greatly improved. The major criticism of including sufficient details in the methodology and results has been largely addressed but still suffers from very poor detail in the Chip-seq/enhancer/database area (more below). It could do with a further review of grammatical errors (e.g. line 133, 199, 214 etc.) and structure to make it more accessible, but this is a minor point.

Experimental design

Chip-seq data is poorly described - see more in the validity of findings section.

Validity of the findings

The use of enhancers in the manuscript is still confusing but the responses to my and reviewer two’s questions have guided what the author has actually done. I now do not believe that Mesp1 has been used in any way, or the author is still not making this clear. Instead this is an assumption and merely an observation of when Mesp is expected to be expressed and needs to be revised to reflect this. Instead, it appears to be an analysis that is independent of Mesp1. Indeed in the author’s response they state that Mesp progenitor data will be added in the future. Line 449-451, states that “In the present definition…” – this is not adequate and I don’t feel this truly reflects the analysis. This then points to Reviewer two concerns about the ChiP-seq description/enhancer collation. The sources of the data needs to be highlighted better as well as a precise definition for inclusion. This is also the case for the RNA-seq data. This would be easily addressed by inclusion of a supplementary table listing the actual datasets used. E.g. which GSMXXXX from GSE29184 for instance and others that were used specifically in the analysis. Figure S3 – suggests 7 datasets for the RNA-seq. I constantly find the database of enhancers, its actually utility and the assumptions that have gone into this a distraction from the main result of this paper. This needs to be clearly articulate/justified.

---

## Round 0.4 · accepted · Accept

Thank you for responding to the reviewer's comments. We think this has made the manuscript better.